# A Proteomic Approach for Understanding the Mechanisms of Delayed Corneal Wound Healing in Diabetic Keratopathy Using Diabetic Model Rat

**DOI:** 10.3390/ijms19113635

**Published:** 2018-11-18

**Authors:** Tetsushi Yamamoto, Hiroko Otake, Noriko Hiramatsu, Naoki Yamamoto, Atsushi Taga, Noriaki Nagai

**Affiliations:** 1Faculty of Pharmacy, Kindai University, Higashi-Osaka 577-8502, Japan; yamatetsu@phar.kindai.ac.jp (T.Y.); hotake@phar.kindai.ac.jp (H.O.); punk@phar.kindai.ac.jp (A.T.); 2Laboratory of Molecularbiology and Histochemistry, Fujita Health University Institute of Joint Research, 1-98 Dengakugakubo, Kutsukake, Toyoake, Aichi 470-1192, Japan; norikoh@fujita-hu.ac.jp (N.H.); naokiy@fujita-hu.ac.jp (N.Y.)

**Keywords:** lumican, diabetic keratopathy, shotgun proteomics, corneal wound healing streptozotocin-induced diabetic rat

## Abstract

Diabetes mellitus is a widespread metabolic disorder, and long-term hyperglycemia in diabetics leads to diabetic keratopathy. In the present study, we used a shotgun liquid chromatography/mass spectrometry-based global proteomic approach using the cornea of streptozotocin-induced diabetic (STZ) rats to examine the mechanisms of delayed corneal wound healing in diabetic keratopathy. Applying a label-free quantitation method based on spectral counting, we identified 188 proteins that showed expression changes of >2.0-fold in the cornea of STZ rats. In particular, the level of lumican expression in the cornea of STZ rats was higher than that of the normal rats. In the cornea of the normal rat, the expression level of lumican was elevated during the wound healing process, and it returned to the same expression level as before cornea injury after the wound was healed completely. On the other hand, a high expression level of lumican in the cornea of STZ rats was still maintained even after the wound was healed completely. In addition, adhesion deficiency in corneal basal cells and Bowman’s membrane was observed in the STZ rat. Thus, abnormally overexpressed lumican may lead to adhesion deficiency in the cornea of STZ rats.

## 1. Introduction

Diabetes mellitus (DM) is a widespread metabolic disorder. The world prevalence of DM among adults (aged 20 to 79 years old) in 2010 was 6.4% (285 million adults), and will increase to 7.7% (439 million adults) by 2030 [1]. DM is characterized by insulin resistance and impaired insulin secretion [2], and the resulting long-term hyperglycemia in diabetics leads to many ophthalmic complications, including retinopathy, cataracts, uveitis, and keratopathy [3]. Patients with ophthalmic complications caused by DM may develop impaired vision or blindness. Diabetic keratopathy shows relatively few symptoms compared with other ophthalmic complications of DM; however, once the cornea is damaged, delayed corneal wound healing is often observed [4]. The delayed corneal wound healing in diabetic keratopathy is caused by changes in the structure and function of the corneal epithelium and the corneal epithelial basement membrane at the time of healing [5,6,7,8,9,10,11,12,13,14,15]. During corneal wound healing, various specific proteins are upregulated, such as vinculin [16], keratins [17], CD44 hyaluronan receptors [18], gelatinases, and metalloproteinase inhibitors [19,20]. Changes in the expression of these proteins are key factors in the process of corneal wound healing in diabetic keratopathy due to the modulation of cell adhesion or migration in the cornea. In addition, the extracellular matrix (ECM) in the cornea (consisting of glycoproteins and proteoglycan) is essential for the maintenance of corneal transparency, and changes in the proteoglycan content of the cornea may result in corneal opacity [21,22,23]. In particular, lumican, is a member of the class II small leucine-rich proteoglycan (SLRP) family [24,25] and is considered as one of the key factors in the corneal wound healing process. Saika et al. reported that lumican has the characteristic of modulating the corneal wound healing of epithelial defects which is consistent with an emerging body of data showing that SLRP proteins not only serve as regulators of collagenous extracellular matrix assembly, but also have biological roles involving direct interactions with cells [26]. In addition, Chakravarti et al. reported that mice lacking lumican show age-dependent corneal opacity and a high proportion of abnormally thick collagen fibers in the corneal stroma [27]. Therefore, changes in the constituents of ECM including lumican are also an important factor during corneal wound healing in diabetic keratopathy because of the need to maintain the normal structure of the cornea. On the other hand, not all of the underlying mechanisms for diabetic keratopathy have been revealed, and further investigations are required.

A proteomic approach is used to examine the mechanisms of cell behavior including cornea tissue engineering [28]. In particular, a shotgun liquid chromatography (LC)/mass spectrometry (MS)-based global proteomic analysis method is considered a useful technique [23,29,30]. As we reported previously, changes in protein expression levels in the lenses of streptozotocin-induced diabetic rat (STZ rats) were evaluated using this methodology, and revealed a decrease in superoxide dismutase in the lenses of STZ rats which contributed to the progression of diabetic cataracts [31]. The analysis may be used to elucidate the causal factor of diabetic keratopathy. In the present study, we performed a shotgun proteomic analysis of the cornea of STZ rats to examine the mechanisms of delayed corneal wound healing in diabetic keratopathy. We further investigated whether any of these proteins might be involved in corneal wound healing in STZ rats.

## 2. Results

### 2.1. Identification and Semiquantitative Comparisons of Differentially Expressed Proteins in the Cornea of Normal and STZ Rats

To investigate mechanisms of diabetic keratopathy, we used shotgun proteomics to examine the molecular profiles of proteins that were regulated in the cornea of the diabetic rat model. Applying our search parameters, we identified 276 proteins in the cornea of the normal rat (Normal), and 260 proteins in the cornea of the STZ rat (STZ) (Figure 1). A total of 403 proteins were identified from the cornea of the rat, of which 133 (33.0%) were present in both Normal and STZ, 143 (35.5%) were unique to Normal, and 127 (31.5%) were unique to STZ (Figure 1).

Furthermore, we evaluated the proteins expressed in the cornea of rats using a label-free semiquantitative method based on spectral counting. Figure 2 shows the Rsc values for the proteins identified in the Normal and STZ groups. A positive Rsc value indicated increased expression in the cornea of diabetic rat, and a negative value indicated reduced expression in the cornea of diabetic rat (light gray area). In addition, the normalized spectral abundance factor (NSAF) value for each protein identified in the Normal and STZ groups was also calculated. Proteins with a high positive or negative Rsc value were considered candidate proteins that were regulated in the cornea of diabetic rat. Table 1 shows the proteins differentially expressed (≥2-fold) in the cornea between normal and STZ rats, and this semiquantitative procedure resulted in the identification of 188 proteins that were differentially expressed in the cornea of the diabetic rat (Appendix A). The expression levels of housekeeping proteins, such as β-actin, glyceraldehyde-3-phosphate dehydrogenase and histone H4 did not change in the cornea of the normal and STZ rats (Figure 2).

### 2.2. Functional Annotation of Proteins Regulated by Hyperglycemia

We performed a Gene Ontology (GO) analysis of the candidate proteins that were regulated in the cornea of diabetic rat. We searched for GO terms related to the “cellular component” (Figure 3), to “molecular function” (Appendix A), to “biological processes” (Appendix A) and “Pathway” (Appendix A) in Database for Annotation, Visualization, and Integrated Discovery (DAVID), and focused on the 15 proteins classified as “extracellular matrix”—which plays important roles in cornea wound healing-in GO cellular component terms (Table 1). Among the 15 proteins, we focused on the effect of enhanced expression of lumican in the cornea of STZ since lumican is a one of the key factors in the wound healing process of the cornea.

Fifteen proteins were differentially expressed in the cornea of STZ rat, and were classified as extracellular matrix proteins by gene ontology analysis.

### 2.3. Effects of Hyperglycemia on Lumican Expression in Cornea of Rat

We examined the expression level of lumican protein in the cornea of STZ rat and normal rat to validate the results of the proteomic analysis. Based on its amino acid sequence, lumican has four potential sites for N-linked glycosylation [32,33], and lumican has forms with various molecular weights for different glycosylation patterns [34]. In the cornea, lumican is expressed as the keratin sulfate proteoglycan, and is considered to possess the heterogeneous glycosaminoglycan [32,35]. Previously, various molecular weights of lumican between 37 and 250 kDa due to differences in glycosylation were detected from cell extracts by Western blot [36,37,38,39,40]. Our results also showed that various molecular weights of lumican between 37 and 250 kDa were observed in the cornea of rat, and increased lumican expression was observed in the cornea of the STZ rat (STZ) as compared to in that of the normal rat (Normal) (Figure 4).

### 2.4. Changes in Expression of Lumican during Corneal Wound Healing in STZ Rats

We previously measured the corneal wound healing rate in normal rats with corneal epithelial abrasion [41] and observed changes in cell migration and cell proliferation in the corneal epithelium was observed 6–12 h and 18–25 h after corneal epithelial abrasion [41]. We thus determined the measurement points (14 h and 24 h after corneal epithelial abrasion) to evaluate the expression of lumican during corneal wound healing in STZ rats. The corneal wounds were almost completely healed at 25 h, and corneal wound healing was similar for both normal and STZ rats at all the tested time points after corneal epithelial abrasion as measured using fluorescein and a TRC-50X. No differences in wounds in the corneal surface were evident with fluorescein (Figure 5). 

On the other hand, adhesion deficiency in the corneal basal cells and Bowman’s membrane of STZ rats was observed as compared to in those of normal rats based on histopathological analysis by hematoxylin and eosin (HE) staining (Figure 6, arrowheads).

We then examined the changes in expression of lumican protein in the cornea of normal and STZ rats during the wound healing process. In the cornea of normal rats, the expression of lumican at 14 h increased whereas by 25 h it had returned to the same level as that before abrasion of the corneal epithelium (Figure 7, Normal). The expression of lumican at 14 h also increased in the cornea of STZ rats; however, by 25 h, its expression was still elevated as compared to that before abrasion of the corneal epithelium (Figure 7, STZ).

## 3. Discussion

In a study to elucidate the mechanism of diabetic keratopathy, the selection of the model animal is important. STZ damages pancreatic β-cells in rats, leading to deficient insulin secretion [42,43]. The continuous hyperglycemia state of the STZ rat tends to cause corneal wounds. Therefore, STZ rats have been widely used as a normal diabetic keratopathy model [44]. First, we examined the plasma of STZ rats to investigate the glucose and insulin levels in STZ rat plasma. The plasma glucose levels of STZ rats were significantly higher than those of the normal rats (normal: 82.1 ± 4.0 mg/dL, STZ: 252 ± 9.7 mg/dL, *n* = 8). Furthermore, the plasma insulin level of STZ rats was not detected and the body weight was significantly decreased compared with that of the normal rats (normal: 319.6 ± 9.1 g, STZ: 216.8 ± 11.0 g, *n* = 8). These results showed that the STZ rats developed diabetes mellitus with hyperglycemia and hypoinsulinemia.

In this study, LC/MS-based proteomics analysis was used to elucidate the mechanism of the delay in corneal wound healing in diabetic keratopathy using this diabetic model rat. Although a quantitative value obtained using spectral counting may not be accurate, this value well reflects differences in expression level and has been used in previous studies investigating novel diagnostic biomarkers [29,45,46,47,48]. A total of 188 proteins that showed >2-fold changes in expression were successfully identified in the cornea of the STZ rat (Table 1). We examined the roles of these identified proteins with GO analysis, and then focused on the functions of proteins classified as “extracellular matrix” since these proteins play important roles in supporting wound healing processes. We also focused on lumican, which is a member of this category. Then, we performed Western blot analysis to validate our spectral counting results, and determine whether hyperglycemia led to increased lumican expression in the cornea of rats (Figure 4).

Lumican is a secreted collagen-binding ECM protein of the corneal, dermis, and tendon stroma, arterial wall, and intestinal submucosa [49,50,51,52]. Lumican can interact with various cellular components such as cytokines, growth factors, and cell surface receptors to regulate cell adhesion, proliferation, and migration [24,34]. Moreover, lumican is considered an important phenotypic marker of corneal keratocytes [40,53,54]; it is transiently expressed during tissue repair in murine corneal epithelium and its absence delays epithelial wound healing in vivo [26]. Therefore, study of the lumican expression level may have therapeutic implications for corneal wound healing, and we investigated whether the abnormally overexpressed lumican in the cornea of the diabetic rat affects the corneal wound healing process. However, there was no difference in the wound healing process between normal and STZ rats (Figure 5). Based on these results, Benaouda conducted a study that showed that in the STZ rats, the corneal epithelium wound was completely covered with proliferating epithelium so as to be structurally indistinguishable from the surrounding residual epithelium [55]. However, abnormalities such as epithelial cell thickening and undulations in the Bowman membrane were observed. From this report, we considered that modulating cell behaviors, such as adhesion, migration, and proliferation balance of the STZ rat were the sources of the abnormality in the corneal epithelium wound healing process.

On the other hand, adhesion deficiency in the corneal basal cells and Bowman’s membrane was observed in the STZ rat after the wound was healed completely while there were no structural changes in the normal rat (Figure 6). Lumican is a critical ECM multifunctional effector with a role in regulating collagen assembly and angiogenesis [27,56,57]. Therefore, abnormally overexpressed lumican might lead to these structural changes in the corneas of STZ rats. In addition, the expression level of lumican in the cornea of normal rat was elevated during the wound healing process, and it returned to a level of expression similar to that before cornea injury once the wound had healed completely. A previous report also showed similar changes in the expression of lumican during the cornea wound healing process [26]. In this report, lumican appeared in the migrating epithelium at 1 day after injury, peaked at days 2 and 3, and completely disappeared by day 7 from the immunohistochemical analysis; anti-lumican antibody delayed corneal injury in cultured mouse eyes. Therefore, this expression cycle of lumican was considered to play an important role in the wound healing process. However, the high expression level of lumican in the cornea of STZ rats was still maintained even after the wound was healed completely (Figure 7). To clarify the mechanism of this maintained high expression of lumican, further studies are necessary such as immunohistochemical analysis using the corneal tissue of STZ rats during the wound healing process. In order to improve delayed wound healing in diabetic keratopathy, we considered it important to adjust this abnormally overexpressed lumican to a normal expression level.

Furthermore, expression levels of decorin, biglycan, and prolargin, which also belong to the family of SLRP, were highly expressed in the cornea of diabetic rat. In a recent study, ECM proteins including lumican, decorin, and biglycan were reported to be involved in corneal wound healing, and these agents have the biological function of modulating cell behavior, such as adhesion, migration, and proliferation [26,58,59,60]. In addition, these ECM proteins function as regulators of tissue hydration and collagen fibrillogenesis [44,61,62]. In earlier study, decorin, byglycan, and prolargin were also found to be involved in the corneal wound healing [58,60]. Therefore, further studies are necessary to investigate the correlation between the expression of other ECM proteins and the wound healing process in diabetic keratopathy. In the future, we will investigate the correlation of other components of the extracellular matrix.

## 4. Materials and Methods

### 4.1. Materials

We purchased urea from GE Healthcare UK Ltd. (Buckinghamshire, UK) and thiourea and Triton X-100 from NACALAI TESQUE Inc. (Kyoto, Japan). All the other reagents and solvents used were of analytical or HPLC grade.

### 4.2. Animals

Healthy Male Wistar rats were provided by Kiwa Laboratory Animals Co., Ltd. (Wakayama, Japan) and the 6-week old rats were injected with STZ for 2 days (100 mg/kg/day, i.p.) (28), and housed for 2 weeks. The STZ rats with over 200 mg/dL glucose levels were used in this study. Sodium pentobarbital overdose (i.p.) was used to euthanize the rats, and the corneas were excised. The removed corneas were used for the analysis of the corneal wound healing rate (*n* = 15), Western blot (*n* = 12) and hematoxylin and Eosin staining (*n* = 8), and proteomics (*n* = 5). All of the experiments were performed in compliance with the regulations approved by the Ethics Committee of the Kindai University Faculty of Pharmacy (Ethic Committee approval code: KAPS-25-001, Date: 1 April 2013). The rats were housed in a room at 25 °C under a 12-h light–dark cycle (2–3 rats/cage). All rats had access to food and water ad libitum.

### 4.3. Assay of Glucose and Insulin

Blood (50 μL) was sampled without anesthesia from the tail vein of each rat after fasting for 12 h (10:00 a.m.). The plasma glucose level was measured using an Accutrend GCT (Roche Diagnostics GmbH, Mannheim, Germany), and plasma insulin levels were assayed using an ELISA Insulin kit (cat. no. M1103; Morinaga Institute of Biological Science, Inc., Kanagawa, Japan) according to the manufacturer’s protocol [63]. The dynamic range of the ELISA Insulin kit is 0.1 to 6.4 ng/mL.

### 4.4. Tryptic Digestion of Protein Extracted from the Cornea of Normal and STZ Rats

The normal and STZ rats were euthanized by injecting sodium pentobarbital. Then, the corneas of normal and STZ rats were removed and homogenized in urea lysis buffer (7 M urea, 2 M thiourea, 5% CHAPS, and 1% Triton X-100). The protein concentration was measured using a Bio-Rad Protein Assay kit (cat. no. 5000006JA;Bio-Rad Laboratories, Inc., Hercules, CA, USA). Then, a gel-free trypsin treatment was performed in accordance with the protocol described by Bluemlein and Ralser [64]. Briefly, 10 μg of protein extract from each sample was reduced at 37 °C for 30 min by the addition of 45 mM dithiothreitol and 20 mM tris (2-carboxyethyl) phosphine in 50 mM ammonium bicarbonate buffer. The proteins were then alkylated with 100 mM iodoacetic acid in 50 mM ammonium bicarbonate buffer at 37 °C for 30 min. After alkylation, the samples were digested at 37 °C for 24 h using mass spectrometry grade trypsin gold (Promega Corp., Madison, WI, USA) at a trypsin/protein ratio of 1:100 (*w*:*w*). Finally, the digests were purified using PepClean C-18 Spin Columns (cat. no. 89870; Thermo, Rockford, IL, USA) following the manufacturer’s protocol.

### 4.5. LC-MS/MS Analysis for Protein Identification

Peptide samples (~2 μg) were injected into a peptide L-trap column (Chemicals Evaluation and Research Institute, Tokyo, Japan) with an HTC PAL autosampler (CTC Analytics, Zwingen, Switzerland). They were then further separated through a Paradigm MS4 (AMR Inc., Tokyo, Japan) with a reverse-phase C18-column (L-column, 3-μm-diameter gel particles, 120 Å pore size, 0.2 × 150 mm; Chemicals Evaluation and Research Institute). The column flow rate was 1 μL/min, and the mobile phase consisted of 0.1% formic acid in water (solution A) and acetonitrile (solution B), with a concentration gradient of 5% solution B to 40% solution B over 120 min. Gradient-eluted peptides were analyzed with an LTQ ion-trap mass spectrometer (Thermo). The results were acquired in a data-dependent manner, with MS/MS fragmentation performed on the two most intense peaks of every full MS scan with an isolation width of *m*/*z* 1.0 and a collisional activation amplitude of 35% in the *m*/*z* rage of 300 to 2000. All MS/MS spectral data were searched against the SwissProt Rattus database using Mascot version 2.4.01 (Matrix Science, London, UK). The search criteria were enzyme as trypsin, with the following allowances: ≤2 missed cleavage peptides; mass tolerance, ±2.0 Da; MS/MS tolerance, ±0.8 Da; cysteine carbamidomethylation for fixed modification; and methionine oxidation modifications for variable modification.

### 4.6. Semiquantitative Analysis of Identified Proteins

The fold change in expression was calculated as the log_2_ of the ratio of protein abundances (Rsc) evaluated by spectral counting [65]. Rsc was calculated using the following equation.
Rsc = log_2_[(n_s_ + f)/(n_n_ + f)] + log_2_[(t_n_ − n_n_ + f)/(t_s_ − n_s_ + f)](1)

Here, n_n_ and n_s_ are spectral counts for the protein in the cornea of normal rat and STZ rat, respectively, t_n_ and t_s_ are total numbers of spectra over all proteins in the two samples, and ƒ is a correction factor set to 1.25.

For comparison, we calculated the relative amounts of identified proteins using the normalized spectral abundance factor (NSAF) [66]. NSAF was calculated using the following equation.
NSAF = (SpC_n_/L_n_)/SUM(SpC_n_/L_n_)(2)

Here, SpC_n_ is the spectral count for the protein in the cornea of normal rat or STZ rat, and L_n_ is the length of the protein in the cornea of normal rat or STZ rat.

Differentially expressed proteins were selected when the Rsc was >1 or <−1, which corresponded to fold changes of >2 or <0.5.

### 4.7. Bioinformatics

We additionally investigated the functions of proteins that showed significantly altered expression with diabetes mellitus. Their sequences were processed by examining their functional annotations in the Database for Annotation, Visualization, and Integrated Discovery (DAVID) version 6.8 (http://david.abcc.ncifcrf.gov/home.jsp) [67,68,69].

### 4.8. Western Blot Analysis

An aliquot of cell extract (10 µg) was added to each well, and subjected to SDS-PAGE under reducing conditions. The separated proteins were then transferred to polyvinylidene fluoride membranes for 30 min at 15 V. After blocking in TBS-Tween-20 (0.1%) buffer with 5% skim milk for 2 h at room temperature, the membranes were incubated at 4 °C overnight with an anti-lumican antibody (1:1000; cat. no. MAB2846; R&D Systems, Inc., Minneapolis, MN, USA). Next, the membranes were washed and incubated at room temperature for 1 h with HRP-conjugated anti-mouse IgG antibody (cat. no. A102PU; American Qualex, San Clemente, CA, USA). The blots were washed and visualized using SuperSignal West Dura Extended Duration substrate (Thermo Fisher Scientific, Inc., Rockford, IL, USA), with band detection using the myECL Imager system (version 2.0; Thermo Fisher Scientific). Finally, the same membranes were re-probed using an anti-β-actin antibody (cat. no. sc-47778; Santa Cruz Biotechnology, Inc., Dallas, TX, USA), which served as the protein loading control. All Western blots were performed in three independent experiments.

### 4.9. In Vivo Wound Healing of the Corneal Epithelium of the Rats

The abrasion of the corneal epithelium in the normal and STZ rats was performed as described previously [41,55,70,71,72,73]. The rats were anesthetized with sodium pentobarbital (30 mg/kg, i.p.), and a 3.0 mm diameter circle was outlined in the center of the cornea with a disposable dermatological skin punch (BIOPSY PUNCH, Kai Industries Co., Ltd., Gifu, Japan). The encircled corneal epithelia were removed with a BD Micro-SharpTM (blade 3.5 mm, 30°, Becton Dickinson, Fukushima, Japan). The corneal area, from which the epithelium was removed, was dyed by instilling a solution containing 1% fluorescein (Alcon, Tokyo, Japan) and 0.4% Benoxil (Santen Pharmaceutical Co., Ltd., Osaka, Japan) [74]. Changes in the corneal wound area were monitored using a TRC-50X (Topcon, Tokyo, Japan) equipped with a digital camera (EOS Kiss Digital N, Canon Inc., Tokyo, Japan) [41] 6 h and 25 h after the corneal epithelial abrasion; the wound area was analyzed with image analyzing software Image J. Corneal wound healing (%) was calculated using the following equation.
Corneal wound healing (%) = (wound area_0 h_ − wound area_25 h_)/wound area_0 h_ × 100(3)

### 4.10. Hematoxylin and Eosin Staining of the Cornea

Hematoxylin and eosin (HE) staining of the cornea in the normal and STZ rats was prepared as described previously [44,75,76]. The rats were anesthetized with sodium pentobarbital (30 mg/kg, i.p.) and the eyes were removed. The removed eyes were fixed in SUPER FIX^TM^ rapid fixative solution (cat. no. KY-500; Kurabo Industries, Osaka, Japan), and 3-μm paraffin sections were prepared and stained with H.E. A microscope (Power BX-51, Olympus, Tokyo, Japan) was used for observation of the corneal tissue.

### 4.11. Statistical Analysis

All data are presented as mean ± standard error of measurement (SEM). Statistical comparisons were made with a one-way analysis of variance (ANOVA). Comparisons of means were performed using Student’s *t*-test (two tail), and the significance level was set at *p* < 0.05.

## 5. Conclusions

In this study, shotgun LC/MS-based global proteomic analysis revealed that a higher expression level of lumican was induced in the cornea of STZ rats, and structural changes in the cornea were observed in the STZ rats after abrasion of the corneal epithelium. Therefore, regulating the lumican expression may reduce the delay in wound healing in diabetic keratopathy and may serve an important role in therapeutic approaches to diabetic keratopathy.

## Figures and Tables

**Figure 1 ijms-19-03635-f001:**
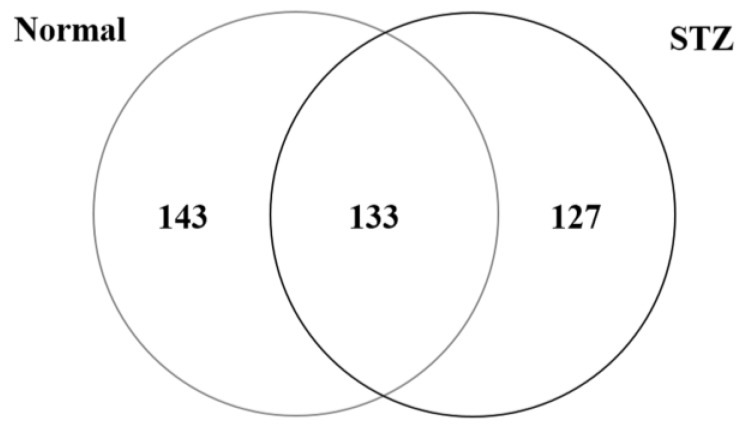
A Venn map of proteins identified from the cornea of normal and streptozotocin (STZ)-induced diabetic rats. We identified 276 proteins in the cornea of normal rats (Normal) and 260 proteins in cornea of STZ-induced diabetic rats (STZ).

**Figure 2 ijms-19-03635-f002:**
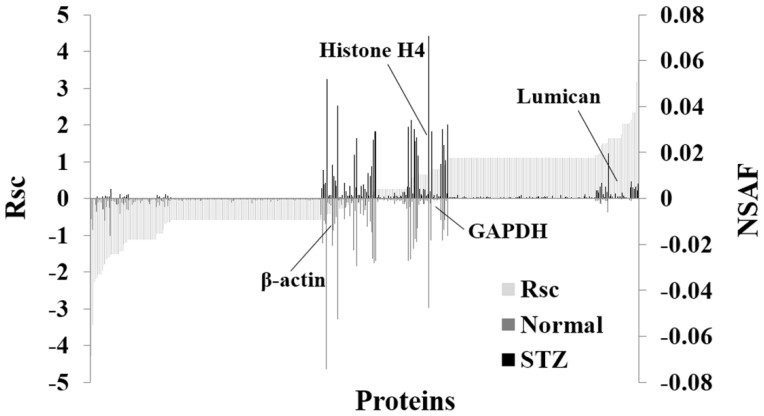
Semiquantitative comparison of proteins differentially expressed in the cornea of the STZ rat. For the identified proteins, Rsc and normalized spectral abundance factor (NSAF) values were calculated to compare protein expression levels between the cornea of normal rat (Normal) and that of the STZ rat (STZ). Proteins are positioned along the x-axis according to Rsc value, increasing from left to right (light gray area). NSAF values are shown for the Normal group (below the axis; gray bar) and for the STZ group (above the axis; black bar). Proteins highly expressed in the Normal and STZ groups are near the left and right sides, respectively of the x-axis. Housekeeping proteins are located around the center of the x-axis.

**Figure 3 ijms-19-03635-f003:**
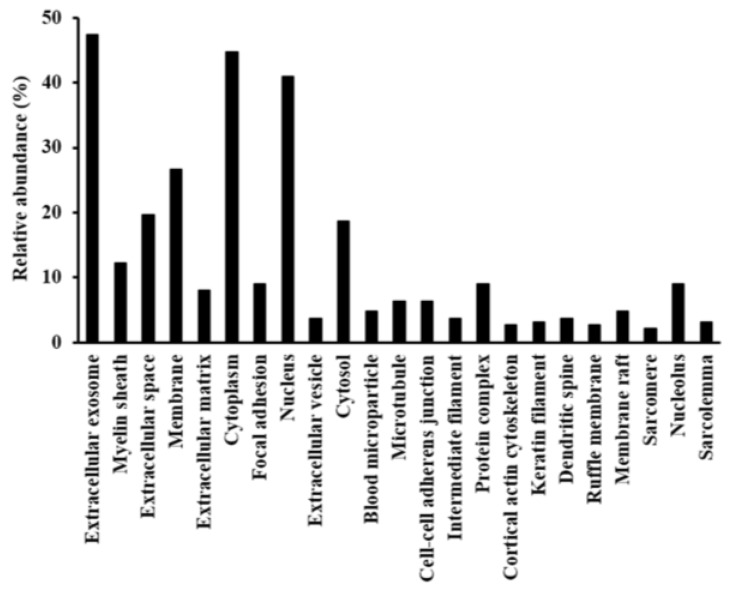
Gene Ontology (GO) analysis of identified proteins. Differentially expressed proteins were assigned to GO term categories related to cellular components, and only significant categories are shown (*p* < 0.05).

**Figure 4 ijms-19-03635-f004:**
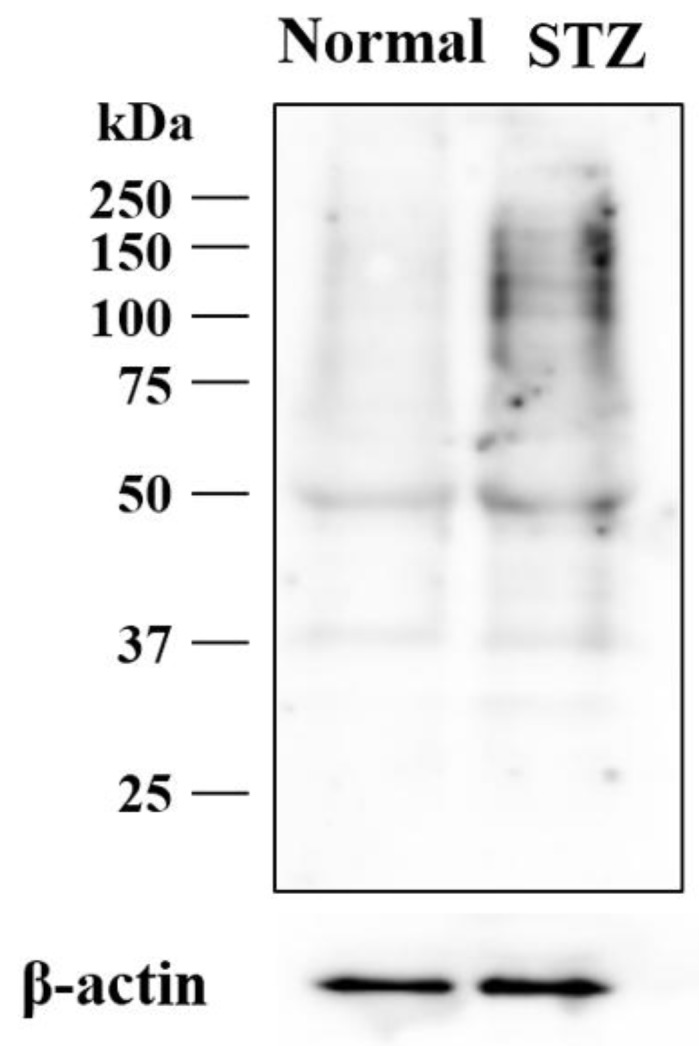
Lumican expression in the cornea of STZ rat. The lumican expression level in the cornea of STZ rats was higher than in that of the normal rats.

**Figure 5 ijms-19-03635-f005:**
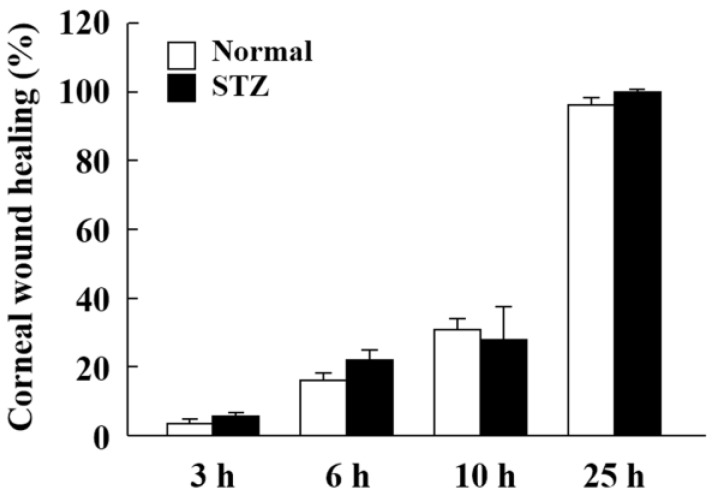
The corneal wound healing of normal and STZ rats after corneal epithelial abrasion. The corneal wound healing values (%) were calculated according to Equation 3. The data are presented as means ± S.E. of 5–10 rats.

**Figure 6 ijms-19-03635-f006:**
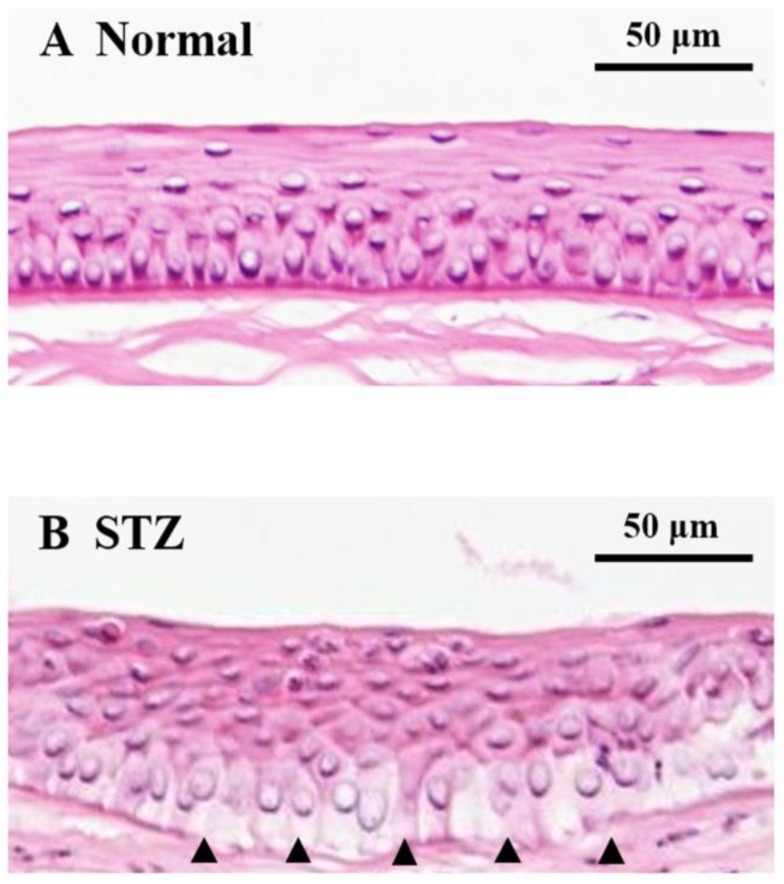
Hematoxylin and eosin staining of the cornea in normal and STZ rats 25 h after corneal wound abrasion. (**A**) Corneal epithelial image of normal rats. (**B**) Corneal epithelial image of STZ rats. Arrow (▲) shows adhesion deficiency between corneal basal cells and Bowman’s membrane.

**Figure 7 ijms-19-03635-f007:**
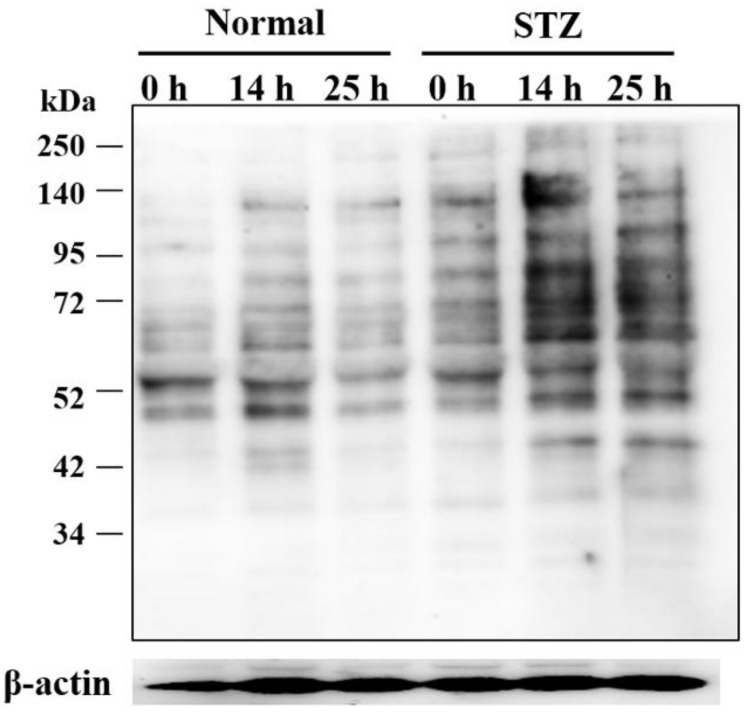
Changes in expression of lumican during the corneal wound healing process. The expression of lumican increased at 14 h in both normal and STZ rats. The high expression level of lumican was still observed in the cornea of STZ rats 25 h after corneal abrasion. β-actin was used as control.

**Table 1 ijms-19-03635-t001:** Proteins categorized as “extracellular matrix” in Gene Ontology analysis.

No.	Accession Number and Description		Fold Change (Rsc)
1	P30427	Plectin	−2.277
2	P69897	Tubulin beta-5 chain	−2.154
3	Q6P9T8	Tubulin beta-4B chain	−2.062
4	P05197	Elongation factor 2	−2.062
5	P61980	Heterogeneous nuclear ribonucleoprotein K	−1.505
6	P61751	ADP-ribosylation factor 4	−1.505
7	P62271	40S ribosomal protein S18	−1.117
8	P45592	Cofilin-1	−1.117
9	P49744	Thrombospondin-4	1.111
10	P02466	Collagen alpha-2(I) chain	1.111
11	Q8CJD3	Zymogen granule membrane protein 16	1.111
12	P51886	Lumican	1.459
13	Q9EQP5	Prolargin	1.488
14	P47853	Biglycan	1.740
15	Q01129	Decorin	2.0601

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
