# Peer review of "A Proteomic Approach for Understanding the Mechanisms of Delayed Corneal Wound Healing in Diabetic Keratopathy Using Diabetic Model Rat"

_ijms, 2018, doi:10.3390/ijms19113635_

Round 1
Reviewer 1 Report
Yamamoto. et.al., were evaluated the abnormal protein expression in delayed corneal wound healing in diabetic keratopathy using a shotgun LC/MS-based global proteomic analysis. The manuscript has an interesting observation that lumican an extracellular matrix protein expression is high in the cornea of streptozotocin induced diabetic rats compared to control animals.
Though the design of the study is appropriate enough to achieve the aim mentioned in manuscript, the current form of manuscript is very poorly presented and the authors are advised to edit the manuscript professionally before submission.
Some imperfections listed below:
1. Please check the subheadings 4.9 and 4.11 and replace 4.11 with ‘Statistics’
2. Authors are advised to include reagent catalog numbers (ex: Lumican antibody) used in this study.
3. Line 108: prosecco??
4. Line 169: diabetic mellitus?
5. Quality of immunoblots (fig.4 &7) are very poor. Authors are advised to present clear blots and mark the Lumican band as the lanes has many nonspecific bands. What do the authors mean by ‘Our results showed various molecular weight of lumican between 37 to 250 kDa’?
6. Language could be improved in many places (ex: Line164-165, we examined the plasma of STZ rats to investigate of glucose and insulin level in STZ rat’s plasma) especially conclusions.
Author Response
Response to Reviewer 1 Comments
Point 1: Please check the subheadings 4.9 and 4.11 and replace 4.11 with ‘Statistics’
Response 1: We checked the subheadings 4.9 and 4.11 and changed the subheadings 4.11 to “Statistical analysis”.
Point 2: Authors are advised to include reagent catalog numbers (ex: Lumican antibody) used in this study.
Response 2: In response to the reviewer’s comment, we added reagent catalog numbers in “Materials and Methods” section.
Point 3: Line 108: prosecco??
Response 3: We changed the text to “process” at the line 115.
Point 4: Line 169: diabetic mellitus?
Response 4: We changed the text to “diabetes mellitus” at the line 183.
Point 5: Quality of immunoblots (fig.4 &7) are very poor. Authors are advised to present clear blots and mark the Lumican band as the lanes has many nonspecific bands. What do the authors mean by ‘Our results showed various molecular weight of lumican between 37 to 250 kDa’?
Response 5: Thank you very much for pointing this out. Previous reports showed that various molecular weights of lumican between 37 to 250 kDa due to the difference glycosylation were detected from the cell extracts of various cell lines and tissue by western blot as well as our present study. In response to this comment, we added this information and related references in “Results” section.
Point 6: Language could be improved in many places (ex: Line164-165, we examined the plasma of STZ rats to investigate of glucose and insulin level in STZ rat’s plasma) especially conclusions.
Response 6: Our manuscript was proofread by a native English-speaking person with sufficient scientific knowledge prior to re-submission.

Reviewer 2 Report
this is a interesting study. the results are useful in the field.
however, there are a few issues need to be clarified or improved.
Abstract: Line 14, the beginning sentence "Diabetes mellitus is characterized by insulin resistance and impaired insulin secretion...", actually, type 1 diabetes is not caused by insulin resistance, and STZ diabetic rat model is a more closely resemble of type 1 diabetes. Therefore, suggest the author to rewrite this sentence to be more fit with the content of this project.
Line 66. "We identified 188....", in the introduction, no result should be included. Please revise this sentence.
There seems to be a problem with special characters in this pdf file. All special characters seem to be disappearing, such as beta, micro, etc. For example line 91, beta-actin's beta is missing; line 140, the +/- sign is missing; line 157, 162, beta missing; line 231, degree is missing; line 234, micro is missing in front of "L". line 232, "ad libitum" should be italicised; line 316, micro is missing.
Statistical analysis should be more detail. t-test is one tail or two tail? data distribution pre-checked? any transformation done? etc
Author Response
Response to Reviewer 2 Comments
Point 1: Abstract: Line 14, the beginning sentence "Diabetes mellitus is characterized by insulin resistance and impaired insulin secretion...", actually, type 1 diabetes is not caused by insulin resistance, and STZ diabetic rat model is a more closely resemble of type 1 diabetes. Therefore, suggest the author to rewrite this sentence to be more fit with the content of this project.
Response 1: Thank you very much for your helpful comments. In response to this comment, we changed the sentence at line 14.
Point 2: Line 66. "We identified 188....", in the introduction, no result should be included. Please revise this sentence.
Response 2: Thank you very much for your helpful comments. In response to this comment, we removed this sentence.
Point 3: There seems to be a problem with special characters in this pdf file. All special characters seem to be disappearing, such as beta, micro, etc. For example line 91, beta-actin's beta is missing; line 140, the +/- sign is missing; line 157, 162, beta missing; line 231, degree is missing; line 234, micro is missing in front of "L". line 232, "ad libitum" should be italicised; line 316, micro is missing.
Response 3: We checked and corrected problems with special characters in our manuscript.
Point 4: Statistical analysis should be more detail. t-test is one tail or two tail? data distribution pre-checked? any transformation done? etc
Response 4: In response to the reviewer’s comment, we changed the text about statistical analysis in “Materials and Methods” section.

Reviewer 3 Report
The manuscript aims to report a proteomic approach for understanding the mechanisms of delayed corneal wound healing in diabetic keratopathy using diabetic model rat. In my opinion, the scientific content indeed involve molecular sciences and seems suitable for consideration for publication in IJMS if the authors’ improvements are deemed adequate.
Specific comments:
After checking the literature database such as PubMed, this reviewer does not find any publications regarding the combination of keywords “corneal wound healing” and “diabetic keratopathy” and “proteomic” or “diabetic” and “lumican” and “cornea”. Therefore, this manuscript may have sufficient academic novelty. However, this reviewer strongly recommends that the authors should emphasize their research motivation since no hypothesis is tested here. The journal readers are curious about why the authors simply focus on the lumican expression during corneal wound healing although they identified 188 proteins and found a relatively high level of lumican expression in rat cornea. Please tell the audiences the physiological significance and role of lumican expression level in corneal wound healing and add appropriate citation to the reference list to support your claims. Otherwise, it is really difficult to clarify the necessity of investigating changes in lumican expression to facilitate understanding of mechanism of corneal wound healing process. For example: lumican is an important phenotypic marker of corneal keratocytes (please refer to the following paper: Lai, J.-Y.; Tu, I.-H. dhesion, phenotypic expression, and biosynthetic capacity of corneal keratocytes on surfaces coated with hyaluronic acid of different molecular weights. Acta Biomater. 2012, 8, 1068-1079.). Therefore, the study of lumican expression level may have therapeutic implications of corneal wound healing.
In this study, the authors adopt proteomic approach to explore the mechanisms of delayed corneal wound healing in diabetic keratopathy. In order to attract general attention from audiences, the authors should enrich the Introduction section by adding appropriate citations to the reference list to emphasize that the proteomic approach is a useful technique to identify and analyze the proteins that may be involved in the mechanism of cell behaviors for potential corneal tissue engineering applications. Please refer to the following example paper: Ma, D.H.-K.; Lai, J.-Y.; Yu, S.-T.; Liu, J.-Y.; Yang, U.; Chen, H.-C.; Yeh, L.-K.; Ho, Y.-J.; Chang, G.; Wang, S.-F.; Chen, J.-K.; Lin, K.-K. Up-regulation of heat shock protein 70-1 (Hsp70-1) in human limbo-corneal epithelial cells cultivated on amniotic membrane: a proteomic study. J. Cell. Physiol. 2012, 227, 2030-2039.
Figure 2 shows the semi-quantitative comparison of proteins differentially expressed in cornea of the STZ rats. However, this figure is a black-white version. How the authors can distinguish the blue area and green bar in the figure? Please improve.
The authors do not provide Table 2. Please must include the data before resubmission. Furthermore, Table S1 in the Supplementary file should be clearly labeled. Currently, the authors also describe Table S1 as “Table 1 in the supplementary file”. It is misleading to the readers.
Figure 5 shows the results of percentage of corneal wound healing of normal and STZ rats at 6 h and 25 h after corneal epithelial abrasion. However, the authors should also examine the data at other time points within 25 h after corneal epithelial abrasion to more comprehensively explore the mechanism of tissue repair.
Furthermore, why the tissue repair in the authors’ case is achieved within a relatively short time period (i.e., 25 h)?
Figure 6 shows the H-E staining of the cornea in normal and STZ rats after 25 h of corneal wound abrasion. In addition to H-E stained images, the authors should provide immunohistochemical staining data of lumican expression in these tissue specimens to confirm the variation of the protein expression levels. Furthermore, it is necessary to include the control groups after immediate corneal abrasion (0 h) in the in vivo studies to reasonably compare the differences between wounded and repaired animals.
The evidences of monitoring corneal tissue repair are insufficient in this work. As stated by the authors, the corneal area, from which the epithelium was removed, was dyed by instilling a solution containing 1% fluorescein. They should provide the time-course fluorescein staining data of corneal histology to better confirm the level of epithelial recovery. The corneal wound closure should be evidenced by slit-lamp biomicroscopy observations. It is necessary to include these fundamental in vivo data to better support the results of protein expression levels.
Figure 7 shows the changes in expression in lumican during corneal wound healing process. However, the beta-actin protein expression levels are not totally the same among different groups. How the authors can accurately normalize the data of lumican expression based on beta-actin expression level? Please carefully check the data again and improve this important point.
As stated by the authors, this expression cycle of lumican was considered to play an important role in the wound healing process. In my opinion, this important sentence should be supported by citing appropriate literature work. Otherwise, the journal readers cannot understand why the high expression level of lumican in cornea of STZ rats was still maintained even after the wound was cured completely? Furthermore, it is unclear that why the “lumican expression level” will be highly correlated with “delayed wound healing”? Please provide any relevant literature to support such an important claim.
Author Response
Response to Reviewer Comments
Point 1: After checking the literature database such as PubMed, this reviewer does not find any publications regarding the combination of keywords “corneal wound healing” and “diabetic keratopathy” and “proteomic” or “diabetic” and “lumican” and “cornea”. Therefore, this manuscript may have sufficient academic novelty. However, this reviewer strongly recommends that the authors should emphasize their research motivation since no hypothesis is tested here. The journal readers are curious about why the authors simply focus on the lumican expression during corneal wound healing although they identified 188 proteins and found a relatively high level of lumican expression in rat cornea. Please tell the audiences the physiological significance and role of lumican expression level in corneal wound healing and add appropriate citation to the reference list to support your claims. Otherwise, it is really difficult to clarify the necessity of investigating changes in lumican expression to facilitate understanding of mechanism of corneal wound healing process. For example: lumican is an important phenotypic marker of corneal keratocytes (please refer to the following paper: Lai, J.-Y.; Tu, I.-H. dhesion, phenotypic expression, and biosynthetic capacity of corneal keratocytes on surfaces coated with hyaluronic acid of different molecular weights. Acta Biomater. 2012, 8, 1068-1079.). Therefore, the study of lumican expression level may have therapeutic implications of corneal wound healing.
Response 1: Thank you very much for your helpful comments. In response to this comment, we added information about physiological significance and role of lumican expression level in wound healing process and related references including your recommendation in “Discussion” section.
Point 2: In this study, the authors adopt proteomic approach to explore the mechanisms of delayed corneal wound healing in diabetic keratopathy. In order to attract general attention from audiences, the authors should enrich the Introduction section by adding appropriate citations to the reference list to emphasize that the proteomic approach is a useful technique to identify and analyze the proteins that may be involved in the mechanism of cell behaviors for potential corneal tissue engineering applications. Please refer to the following example paper: Ma, D.H.-K.; Lai, J.-Y.; Yu, S.-T.; Liu, J.-Y.; Yang, U.; Chen, H.-C.; Yeh, L.-K.; Ho, Y.-J.; Chang, G.; Wang, S.-F.; Chen, J.-K.; Lin, K.-K. Up-regulation of heat shock protein 70-1 (Hsp70-1) in human limbo-corneal epithelial cells cultivated on amniotic membrane: a proteomic study. J. Cell. Physiol. 2012, 227, 2030-2039.
Response 2: Thank you very much for your helpful comments. In response to this comment, we added information about usefulness of proteome approach to examine the mechanism of cell behaviour and related references including your recommendation in “Introduction” section.
Point 3: Figure 2 shows the semi-quantitative comparison of proteins differentially expressed in cornea of the STZ rats. However, this figure is a black-white version. How the authors can distinguish the blue area and green bar in the figure? Please improve.
Response 3: We changed the text to “light grey area” at the line 89, “light grey” at the line 103, “grey” at the line 104 and “black” at the line 104.
Point 4: The authors do not provide Table 2. Please must include the data before resubmission. Furthermore, Table S1 in the Supplementary file should be clearly labeled. Currently, the authors also describe Table S1 as “Table 1 in the supplementary file”. It is misleading to the readers.
Response 4: We changed the text to “Supplementary Table” at the line 94 and “Table 1” at the line 113.
Point 5: Figure 5 shows the results of percentage of corneal wound healing of normal and STZ rats at 6 h and 25 h after corneal epithelial abrasion. However, the authors should also examine the data at other time points within 25 h after corneal epithelial abrasion to more comprehensively explore the mechanism of tissue repair.
Response 5: In response to the reviewer’s comment, we added data of other time point on cornea wound healing in Fig. 5.
Point 6: Furthermore, why the tissue repair in the authors’ case is achieved within a relatively short time period (i.e., 25 h)?
Response 6: Thank you very much for pointing this out. In this study, we observed tissue repair process until 25 h since corneal wounds were almost completely cured at 25 h as shown in Fig. 5. In response to this comment, we added information about this in “Result” section.
Point 7: Figure 6 shows the H-E staining of the cornea in normal and STZ rats after 25 h of corneal wound abrasion. In addition to H-E stained images, the authors should provide immunohistochemical staining data of lumican expression in these tissue specimens to confirm the variation of the protein expression levels. Furthermore, it is necessary to include the control groups after immediate corneal abrasion (0 h) in the in vivo studies to reasonably compare the differences between wounded and repaired animals.
Response 7: Thank you very much for your helpful comments. Although we think that immunohistochemical analysis of lumican in corneal tissue is important to examine the variation of its expression, in this study, we first examined whether the expression level of lumican was changed between cornea of normal rat and STZ-induced diabetic rat under wound healing process by western blot and showed the high expression level of lumican in cornea of STZ rats was still maintained even after the wound was cured completely. Therefore, we added the significance to perform immunohistochemical analysis of lumican in corneal tissue in the “Discussion” section.
In vivo wound healing model of the corneal epithelium of the rats using in this study was well established method, and previous report already showed the H.E. staining data of wound healing process using cornea of normal rat and STZ-induced diabetic rat. In addition, we also published several reports using this model. We added another references in “Materials and methods” section.
Point 8: The evidences of monitoring corneal tissue repair are insufficient in this work. As stated by the authors, the corneal area, from which the epithelium was removed, was dyed by instilling a solution containing 1% fluorescein. They should provide the time-course fluorescein staining data of corneal histology to better confirm the level of epithelial recovery. The corneal wound closure should be evidenced by slit-lamp biomicroscopy observations. It is necessary to include these fundamental in vivo data to better support the results of protein expression levels.
Response 8: Thank you very much for your helpful comments. In response to this comment, we tried to perform monitoring corneal tissue repair by slit-lamp biomicroscopy observations. However, there is no slit-lamp biomicroscopy in our university. Therefore it is difficult to do it within revised period. On the other hand, we performed the time-course fluorescein staining data of corneal histology and added data of this time course on cornea wound healing in Fig. 5
Point 9: Figure 7 shows the changes in expression in lumican during corneal wound healing process. However, the beta-actin protein expression levels are not totally the same among different groups. How the authors can accurately normalize the data of lumican expression based on beta-actin expression level? Please carefully check the data again and improve this important point.
Response 9: Thank you very much for pointing this out. In response to the reviewer’s comment, we changed western blot data.
Point 10: As stated by the authors, this expression cycle of lumican was considered to play an important role in the wound healing process. In my opinion, this important sentence should be supported by citing appropriate literature work. Otherwise, the journal readers cannot understand why the high expression level of lumican in cornea of STZ rats was still maintained even after the wound was cured completely? Furthermore, it is unclear that why the “lumican expression level” will be highly correlated with “delayed wound healing”? Please provide any relevant literature to support such an important claim.
Response 10: Thank you very much for your helpful comments. In response to this comment, we added information about this comment in “Discussion” section.

Round 2
Reviewer 3 Report
The authors’ revision is highly appreciated. Although they do not follow this reviewer’s suggestions to provide some additional evidences (because of insufficient experimental resources) to support their claim and enrich the article content in the revised version, I respect the authors’ responses and scientific viewpoints.